REGISTERED REPORT PROTOCOL

# Understanding target-specific effects of antidepressant drug pollution on molluscs: A systematic review protocol

**Maurice E. Imiuwa**[1,2]*, **Alice Baynes**[2], **Edwin J. Routledge**[2]*

**1** Faculty of life Sciences, Department of Animal and Environmental Biology, University of Benin, Benin City, Nigeria, **2** College of Health, Medicine and Life Sciences, Brunel University London, Uxbridge, United Kingdom

* eghosamaurice.imiuwa@brunel.ac.uk (MEI); edwin.routledge@brunel.ac.uk (EJR)

## Abstract

### Background

The environmental prevalence of widely prescribed human pharmaceuticals that target key evolutionary conserved biomolecules present across phyla is concerning. Antidepressants, one of the most widely consumed pharmaceuticals globally, have been developed to target biomolecules modulating monoaminergic neurotransmission, thus interfering with the endogenous regulation of multiple key neurophysiological processes. Furthermore, rising prescription and consumption rates of antidepressants caused by the burgeoning incidence of depression is consistent with increasing reports of antidepressant detection in aquatic environments worldwide. Consequently, there are growing concerns that long-term exposure to environmental levels of antidepressants may cause adverse drug target-specific effects on non-target aquatic organisms. While these concerns have resulted in a considerable body of research addressing a range of toxicological endpoints, drug target-specific effects of environmental levels of different classes of antidepressants in non-target aquatic organisms remain to be understood. Interestingly, evidence suggests that molluscs may be more vulnerable to the effects of antidepressants than any other animal phylum, making them invaluable in understanding the effects of antidepressants on wildlife. Here, a protocol for the systematic review of literature to understand drug target-specific effects of environmental levels of different classes of antidepressants on aquatic molluscs is described. The study will provide critical insight needed to understand and characterize effects of antidepressants relevant to regulatory risk assessment decision-making, and/or direct future research efforts.

### Methods

The systematic review will be conducted in line with the guidelines by the Collaboration for Environmental Evidence (CEE). A literature search on Scopus, Web of Science, PubMed, as well as grey literature databases, will be carried out. Using predefined criteria, study selection, critical appraisal and data extraction will be done by multiple reviewers with a web-based evidence synthesis platform. A narrative synthesis of outcomes of selected

This is a Registered Report and may have an associated publication; please check the article page on the journal site for any related articles.

**Data Availability Statement:** All relevant data from this study will be made available upon study completion.

**Funding:** MEI received funding from Tertiary Education Trust Fund (TETFund), Nigeria (https://tetfund.gov.ng/). The funder had no role in study design, data collection and analysis, decision to publish, or preparation of the manuscript.

**Competing interests:** The authors have declared that no competing interests exist.

studies will be presented. The protocol has been registered in the Open Science Framework (OSF) registry with the registration DOI: 10.17605/OSF.IO/P4H8W.

## Introduction

The widespread occurrence of human pharmaceuticals in the environment is a cause of increasing concern. Particularly worrisome is the presence, in the aquatic environment, of neuromodulatory pharmaceuticals developed to specifically target critical-function biomolecules such as monoamine neurotransmitter re-uptake transporters, their synaptic receptors and deamination enzymes in humans, that are well conserved across animal phyla [1–5]. Monoamine neurotransmitters are biogenic amines containing one amine group (and essentially include serotonin, norepinephrine and dopamine), that are critical in the modulation of virtually all brain functions, and play key roles in the regulation of physiological processes such as development, reproduction, autonomic functions, hormone secretion and complex behaviours [6–9]. The synaptic activity of monoamines is tightly modulated by their re-uptake transporters, pre-and post-synaptic receptors and deaminating oxidases [6,10,11]. These critical-function biomolecules regulate the intensity and duration of synaptic monoamine signaling, and for this reason, they are key pharmacological targets for antidepressant drugs [6,12]. Antidepressants are a major class of psychotropic drugs that target and inhibit monoamine re-uptake transporters, their synaptic receptors and terminating enzymes, thereby interfering with the endogenous modulation of monoaminergic neurotransmission, a key neurophysiological process [13]. They are used for the treatment of depression, and are also prescribed for other disorders such as generalized anxiety disorders, obsessive-compulsive disorder, panic disorder, social anxiety disorder and specific phobia [14–16].

Based on their modes of action and chemical structures, antidepressants are classified into four major groups: monoamine oxidase inhibitors (MAOIs), tricyclic antidepressants (TCAs), selective serotonin reuptake inhibitors (SSRIs), and serotonin-norepinephrine reuptake inhibitors (SNRIs) [17,18]. Additionally, however, a fifth group of largely heterogenous antidepressant drugs exists, and are accordingly, referred to as 'atypical' or 'other' antidepressants [19]. Briefly, MAOIs exert inhibitory action on monoamine oxidases (the enzymes that oxidatively deaminate monoamine neurotransmitters). TCAs, named after their tricyclic chemical structures, inhibit serotonin and norepinephrine re-uptake transporters, and also exert antagonistic actions on post-synaptic adrenergic $\alpha_1$ and $\alpha_2$, muscarinic and histamine $H_1$ receptors. SNRIs like TCAs also inhibit serotonin and norepinephrine re-uptake transporters, but they do so with little, or no, pharmacological action on the post-synaptic receptors affected by TCAs. SSRIs act therapeutically as selective inhibitors of serotonin reuptake transporters, while atypical antidepressants exert a range of pharmacological actions on monoamine neurotransmitter system including acting as antagonists and agonists of several pre- and post-synaptic receptors, and inhibitors of serotonin, norepinephrine and dopamine transmembrane transporters [13,19].

Interestingly, in recent times the prescription and consumption of antidepressants have consistently been on the increase due to a burgeoning prevalence of depression in society [20,21]. Indeed, depression is projected to become the leading cause of disease morbidity worldwide by 2030 [20,22,23]. Although readily biotransformed following consumption by patients, antidepressant drugs are essentially excreted as parent compounds and pharmaceutically active metabolites [24–29]. As they are not completely removed by wastewater treatment processes [30,31], antidepressant drugs end up in wastewater effluents discharged into surface

waters [32–34], and in sewage sludge or reclaimed water applied to agricultural land [35]. The fallout of this has been their widespread occurrence and detection in the aquatic environments across the globe, with different antidepressant drugs and their active metabolites detected in soil, ground water, surface water and wildlife [34,36–39]. While current environmental antidepressant levels range from ng/L to low μg/L, they are designed to act on their molecular targets at particularly low concentrations [40,41]. Consequently, investigations into potential effects of exposure to antidepressants in wildlife have been on the increase owing to their known neuromodulatory effects in humans [42,43]. There are now a considerable number of laboratory studies describing a range of toxicological effects following exposure to various antidepressants in different aquatic species. However, drug target-specific effects of different classes of antidepressants in non-target aquatic organisms remain to be understood [18,41,43,44]. Importantly, data suggest molluscs may be more vulnerable to the effects of antidepressants than any other animal phylum because multiple key physiological processes (including reproduction and development) are regulated by monoamines rather than vertebrate-type sex steroids in molluscs. To illustrate, in vertebrates the enzyme 5α-reductase is involved in the conversion of testosterone to the more potent form, dihydrotestosterone (DHT), which is important for the formation of male phenotype and the development of external genitalia during embryogenesis [45–47]. However, inhibition of 5α-reductase during early development in the freshwater pulmonate gastropods, *Biomphalaria glabrata* and *Physella acuta*, has been shown to affect shell formation [48], and to date, no androgen receptor (the target for DHT action) has been identified in molluscs [49–52], while monoamines have been reported to have a role in shell formation in the Pacific oyster, *Crassostrea gigas*. [53,54]. Furthermore, in the bivalves, *Nodipecten subnosus*, *Crassostrea gigas* and *Argopecten purpuratus*, monoamines are detected in the gonads, with increased concentrations during gonadal growth stages, which decrease after spawning [55–57], suggesting a direct role in reproduction. Additionally, in freshwater pulmonate gastropods including *Biomphalaria glabrata*, dopamine is detected in the albumen gland with increased concentrations during perivitelline fluid secretion, while in *Helisoma duryi*, it is involved in perivitelline fluid secretion [58,59]. In *Helisoma trivolvis*, serotonin is involved in larval development via serotonin receptor-modulated cAMP-dependent regulation of cell division [8].

Molluscs are a highly biodiverse group (second only to arthropods in terms of number of species), displaying a wide variety of ecologically unique body forms, sizes, lifestyles, and microhabitat preferences [60,61]. This makes them indispensable for understanding ecological effects of anthropogenic chemicals in the aquatic environments. Molluscan monoamines are produced by the nervous system where they mediate chemical communication between neurons, with other innervated cell types, or exert hormonal action when released into the blood [62–65]. There are also hormone-producing neurons, the neurosecretory cells, which together with their targets, form the neuroendocrine system that is the main source of hormones in molluscs [61,66]. Interestingly, targets for antidepressant action, including monoamine transmembrane transporters, monoamine synaptic receptors and monoamine oxidases, are present in molluscs [67–72], and the effective concentrations of antidepressant drugs in molluscs are in the range of those commonly detected in the aquatic environment [73–75]. Also, the need to consider the effects of substances of high environmental relevance and poor scientific underpinning in molluscs has long been recognized as a priority area [76].

While there are reviews on general effects of antidepressants on aquatic organisms (with data on molluscs), including Fong and Ford [77], published almost 10 years ago; Silva et al. [78], with scope limited to SSRIs; Sehonova et al. [42] and Moreira et al. [79], with very brief sections on molluscan data; and Canesi et al. [80], limited to bivalves, they are all narrative reviews. Based on continued research interest, a considerable number of individual studies on

different pertinent aspects of the subject now exist. The present study, therefore, seeks to understand target-specific effects of environmental levels of different classes of antidepressants in molluscs through a systematic review of literature. The study provides the opportunity to (i) synthesize the first systematic review of the effects of exposure of aquatic molluscs to different classes of antidepressant drugs, with the potential to provide critical insight relevant to regulatory risk assessment decision-making, (ii) identify research gaps in our current understanding of their mechanisms of action, (iii) establish best practice within research studies to improve future work in the field, and (iv) identify questions for which available evidence provide clear answers and further research may not be necessary. The authors are solely responsible for the design and conduct of the study, and it does not involve any form of external organizational stakeholder engagement.

### Objective of the review

**Primary question.**  The primary question of the study is: what are the target-specific effects of environmental levels of different classes of antidepressant drugs on aquatic molluscs? The primary question consists of the following PECO (population, exposure, comparator and outcome) components—Population: molluscs (all aquatic species and all life stages) exposed to laboratory-based water-borne antidepressants; Exposure: acute/chronic exposure to any class of antidepressants and/or their major pharmaceutically active metabolites; Comparator: vehicle-treated or naïve controls; Outcome: all study outcomes directly related to behaviour, movement, feeding, respiration, reproduction, development, immunity, neurophysiology, and intercellular signaling events.

## Methods

The systematic review will be conducted in line with the guidelines by the Collaboration for Environmental Evidence (CEE) [81]. Accordingly, the systematic review protocol was developed following the CEE Reporting standards for Systematic Evidence Synthesis (ROSES) [82] (See S1 Table for ROSES; S2 Table for PRISMA-P in compliance with PLOS One protocol publication criteria). As recommended by Whaley et al. [83], the protocol has been registered in the Open Science Framework (OSF) registry, with the Registration DOI: 10.17605/OSF.IO/P4H8W.

### Searching for articles

While two bibliographic databases are usually considered sufficient for evidence synthesis involving animal studies [84], prioritizing sources with the largest number of relevant articles has been suggested [85]. Consequently, article searches will be conducted in three key bibliographic databases, namely Web of science, Scopus and PubMed. The search strategy outlined in Table 1, comprehensively includes key study population and exposure terms for peer-reviewed original research articles in English language using information retrieval sensitivity and relevance criteria for each of the databases. The search strategy was developed in consultation with an academic liaison librarian as recommended for evidence synthesis [86]. Further, as supplementary searches for grey literature in catalogues of academic theses, databases of conferences and proceedings, preprint servers and funders' databases of on-going research have been recommended for mitigating publication bias in systematic reviews [87], additional searches will be conducted in ProQuest Dissertations and Theses Global, Open Access Theses and Dissertations, OpenGrey, Grey Literature Report, Research square and EcoEvoRxiv for grey literature. The supplementary searches will be carried out using key study terms [88].

**Table 1. Study search strategy for use in Scopus, Web of Science and PubMed.**

| Bibliographic databases | Search strings |
|---|---|
| Scopus | (TITLE-ABS-KEY (mollusc* OR gastropod* OR mussel* OR clam OR clams OR bivalves OR mollusk* OR snail* OR cuttlefish) AND TITLE-ABS-KEY ("psychotropic drug*" OR antidepressants OR sertraline OR fluoxetine OR citalopram OR paroxetine OR amitriptyline OR venlafaxine OR mirtazapine OR dosulepin OR clomipramine OR dosulepin OR escitalopram OR fluvoxamine OR imipramine OR nortriptyline OR lofepramine)) AND (LIMIT-TO (DOCTYPE, "ar")) |
| Web of Science, PubMed | (Mollusc* OR Gastropod* OR Mussel* OR Clam OR clams OR Bivalves OR Mollusk* OR Snail* OR cuttlefish) AND ("psychotropic drug*" OR Antidepressants OR Sertraline OR Fluoxetine OR Citalopram OR paroxetine OR amitriptyline OR venlafaxine OR mirtazapine OR Dosulepin OR Clomipramine OR Dosulepin OR Escitalopram OR Fluvoxamine OR Imipramine OR Nortriptyline OR Lofepramine) |

## Article screening and study eligibility criteria

**Screening process.** A web-based evidence synthesis platform, EPPI-Reviewer [89], will be used to manage article search across all selected databases. In the first phase, articles will be screened using only titles and abstracts within EPPI-Reviewer to facilitate uniform review by two reviewers. The outcome of this screening phase will then be evaluated by both reviewers for correctness, and a third reviewer will be contacted to provide an independent opinion in the event of any discrepancy. In the second phase, the full text of articles that have been selected and approved in the first phase are then screened against the pre-defined inclusion and exclusion criteria shown in Table 2 by two reviewers. In the final phase, full text-screened articles that are selected and approved by the two reviewers using the study PECO-based inclusion and exclusion criteria will be included in the study. This procedure will be replicated for grey literature.

**Table 2. Study inclusion and exclusion criteria.**

| Study parameters | Inclusion criteria | Exclusion criteria |
|---|---|---|
| Study design | Waterborne antidepressant laboratory exposures | *In vitro* studies, feed-borne exposure, injection of antidepressants |
| Population | All genera/species and life stages of aquatic molluscs | Molluscan cell lines, land molluscs, any other animal phylum |
| Exposure | Exposure to all classes of antidepressants (parent compound/active metabolites) singly administered | Antidepressant mixtures, other pharmaceuticals/xenobiotics |
| Outcome measures | Effects on behaviour, locomotion, respiration, feeding, reproduction, development, immunity, neurophysiology, and intracellular signalling events | Any general toxicity effects including biotransformation, cytotoxicity, cytogenetics and mortality |
| Language | English | Any other language |
| Publication date | No restriction | No restriction |
| Others | Nil | Nil |

**Eligibility criteria.** As the study seeks to understand the effects of antidepressants in the aquatic environment on aquatic molluscs, only whole-animal laboratory studies, and not *in vitro* exposure studies, will be included. Additionally, since the pharmaceuticals of interest are antidepressants that are widely detected in the aquatic environment, only waterborne antidepressant exposure studies will be included. As a result, studies on other routes of exposure, including foodborne antidepressants and injection of antidepressants will not be included. On account of the habitat of interest, only aquatic molluscs (all life stages), and not terrestrial species, will be eligible for inclusion. With regard to test chemical eligibility criteria, laboratory exposure studies on all antidepressant drugs and their major active metabolites (since pharmacologically active metabolites of antidepressants are also widely detected in the aquatic environment) will be included. Furthermore, since it is difficult to delineate constituent chemical effects in mixture exposures [90], only studies using singly administered antidepressants will be included. However, studies containing data on both mixtures and singly administered antidepressants will be included in order to extract data for singly administered antidepressants only.

Antidepressants are designed to pharmacologically target monoamine re-uptake transporters, pre-and post-synaptic receptors of monoaminergic neurons, and monoamine oxidases. In molluscs, monoamines have been shown to have functions in key physiological processes including behaviour [91,92], locomotion [93,94], respiration [95,96], feeding [97,98], reproduction [57,59] development [8,99] and immunity [100]. As a result, outcome data directly related to these physiological processes will be included in the study. Furthermore, as recent studies have revealed that in addition to monoaminergic system, other neural targets especially those directly involved in the regulation of neuronal survival, neuronal growth and synaptic plasticity, may play more direct roles in antidepressant effects [101,102], outcome data on neurophysiology, and intracellular signaling events will be included in the study. Conversely, study outcomes other than those selected by these criteria including biotransformation, cytotoxicity, cytogenetics, mortality and any other general toxicity effects will not be included in the review. On the whole, external validity, the relevance of each included study to the systematic review question [103,104], was centrally factored into the eligibility criteria development.

**Study validity assessment.** 'Internal validity', 'risk of bias' or 'critical appraisal' generally describes the quality assessment of each of the included studies in a systematic review [103]. The assessment is usually based on a set of questions defined in advance to address various types of bias [105]. In environmental science, this is generally flexible, and the development and operationalization of specific internal validity assessment tools depend on a number of key study design and performance parameters [104]. Consequently, a comprehensive set of quality parameters bordering on study design and performance were defined for the risk of bias assessment of each included study in this systematic review (Table 3) while adopting a framework of select sources of bias [86,105]. Specifically, our tool is framed into a set of 10 questions which requires a yes-or-no answer. The answers (yes = 1; no = 0) to the quality questions for each of the included studies are summed to further classify each study into any of three quality categories, namely low risk of bias ($\geq 8$), medium risk of bias (6–7) and high risk of bias ($\leq 5$). The appraisal will be done by one reviewer, and evaluated by three reviewers for completeness and consistency.

**Data extraction.** All data on the systematic review PECO statement including study ID (or authors and the year of publication) and data on all study characteristics of each included study will be extracted. Data extraction will be carried out on EPPI-Reviewer platform to facilitate uniform extraction. Extraction will be carried out by one reviewer, while extracted data will be evaluated by two independent reviewers for completeness and consistency. Where there are incomplete data, authors will be contacted for clarifications. Finally, extracted data will be made available as an additional data file.

**Table 3. Study critical appraisal framework.**

| Key study parameter questions | Study ID | | Study ID | | Study ID | |
|---|---|---|---|---|---|---|
| | Yes | No | Yes | No | Yes | No |
| Were the control and treatment groups similar at baseline? | | | | | | |
| Is there any difference in the way the control and treatment groups were handled during the experiment (apart from difference due to treatment)? | | | | | | |
| Was the experiment replicated? | | | | | | |
| Was an appropriate control provided? | | | | | | |
| Were the exposure concentrations experimentally determined in the exposure medium? | | | | | | |
| Is it likely that the water renewal level and frequency are sufficient to maintain exposure conditions? | | | | | | |
| Are the test concentrations environmentally relevant or were the internal (tissue) levels determined? | | | | | | |
| Is there any difference in the way the outcome measures in both control and treatment groups were accessed? | | | | | | |
| Is there selective reporting in the way the outcome measures are presented and reported? | | | | | | |
| Is the study free from any other form of bias of concern not listed here? | | | | | | |
| Summation | ≥ 8 | | 6–7 | | ≤ 5 | |
| Risk of bias level | low risk of bias | | medium risk of bias | | high risk of bias | |

**Potential effect modifiers.** Potential effect modifiers, or factors that may cause some degree of heterogeneity in the response of molluscs exposed to antidepressants, will be extracted from each included study and considered in the review. We have selected the following key potential effect modifiers associated with toxicological responses of biological systems to chemical exposure:

- Species, sex, reproductive strategy, life stage and chosen endpoints

- Antidepressant class and exposure concentrations

- Exposure duration, renewal regime and percentage renewal

## Data synthesis, presentation and discussion

Given the wide variety of aquatic molluscan species, classes of antidepressant drugs and exposure conditions reported in laboratory studies, the data are not considered to be amenable to meta-analysis [105], and only narrative synthesis will be conducted. Accordingly, data on all study characteristics and statistically significant results of each included study will be presented with tables [105]. Further, data within distinct subgroups comprising species of molluscs, class of antidepressants, exposure concentrations and nature of effects will be summarized, compared and contrasted [106]. The synthesis will be followed by an extensive discussion. Where

full text-screened articles are excluded from data synthesis, a list of affected studies with the reason for exclusion will be provided. The EPPI-Centre approach to assessing the overall robustness of the synthesis will be adopted, and described in terms of the internal validity of included studies [107].

## Supporting information

**S1 Table. ROSES.**
(XLSX)

**S2 Table. PRISMA-P.**
(DOCX)

## Acknowledgments

The authors would like to thank Joanne Mcphie, an academic liaison librarian, Brunel University London, for technical assistance with the search strategy development.

## Author Contributions

**Conceptualization:** Maurice E. Imiuwa, Alice Baynes, Edwin J. Routledge.

**Methodology:** Maurice E. Imiuwa.

**Writing – original draft:** Maurice E. Imiuwa.

**Writing – review & editing:** Alice Baynes, Edwin J. Routledge.

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
