## [Decision Letter · Decision Letter 0]

25 Apr 2023

PONE-D-23-08500

Understanding target-specific effects of antidepressant drug pollution on molluscs: a systematic review protocol

PLOS ONE

Dear Dr. IMIUWA,

Thank you for submitting your manuscript to PLOS ONE. After careful consideration, we feel that it has merit but does not fully meet PLOS ONE’s publication criteria as it currently stands. Therefore, we invite you to submit a revised version of the manuscript that addresses the points raised during the review process.

We look forward to receiving your revised manuscript.

Kind regards,

Peter P. Fong, Ph.D

Academic Editor

PLOS ONE

Journal Requirements:

Additional Editor Comments:

Dear Dr. Imiuwa,

Your manuscript has been reviewed by three reviewers and by me. It requires a minor revision. Pay close attention to and respond to the comments of all three reviewers, but especially of reviewers 1 and 3. In particular,

reviewer #1 says that in molluscs, the mode of action is more complicated than simply modulation of monoaminergic pathways and I agree with this. So, it would be appropriate to expand your search to include alternate modes of action

of antidepressants in molluscs. There are published papers on these alternative modes of action.

Reviewers' comments:

Reviewer's Responses to Questions

**Comments to the Author**

1. Does the manuscript provide a valid rationale for the proposed study, with clearly identified and justified research questions?

Reviewer #1: Yes

Reviewer #2: Yes

Reviewer #3: Yes

2. Is the protocol technically sound and planned in a manner that will lead to a meaningful outcome and allow testing the stated hypotheses?

Reviewer #1: Yes

Reviewer #2: Yes

Reviewer #3: Yes

3. Is the methodology feasible and described in sufficient detail to allow the work to be replicable?

Reviewer #1: Yes

Reviewer #2: Yes

Reviewer #3: Yes

4. Have the authors described where all data underlying the findings will be made available when the study is complete?

Reviewer #1: Yes

Reviewer #2: Yes

Reviewer #3: Yes

5. Is the manuscript presented in an intelligible fashion and written in standard English?

Reviewer #1: Yes

Reviewer #2: Yes

Reviewer #3: Yes

6. Review Comments to the Author

You may also provide optional suggestions and comments to authors that they might find helpful in planning their study.

Reviewer #1: The ms. proposes a systematic review protocol for understanding target-specific effects of antidepressants in molluscs. This represents a timely approach in the evaluation of the impact of pharmaceuticals in this invertebrate group.

However, the criteria for inclusion should be better justified. Although the aim of this work is to extract relevant literature data and not descriptive information, the choice of these criteria is extremely narrow, based on the assumption that target-specific effects of antidepressants in molluscs are restricted to the monoaminergic system, which is currently not taken for granted even in humans.

The ms. is clear and well written. Only subheadings are not clear (i.e. Methods and subheadings of the following sections are the same) and should be revised.

117-118 and following paragraph

The statement …. physiological processes (including reproduction and development) are regulated by monoamines rather than vertebrate-type sex steroids in molluscs is obviously correct.

However, in the introduction, a paragraph is lacking underlying the profound differences between the endocrine system in humans as vertebrates with respect to molluscs, where chemical communication is mainly neuroendocrine, and endocrine glands do not exist. For those not familiar with molluscan neuroendocrine system the examples made in paragraph following line 118 are not explanatory enough.

134-135 ecologically unique body forms, sizes, lifestyles, and microhabitat preferences : delete ‘ecological’

139 in the range commonly = in the range of those commonly

139-141 is should be specified that this sentence refers to a 2010 OECD guideline for Mollusc Reproductive

Toxicity Tests in freshwater gastropods

232-36 ….. As a result, only outcome data directly related to these physiological processes and neurophysiology will be included in the study.

We understand the reasons for exclusion criteria as in Table 2.

However, the Authors should consider a wider point of view. Although molluscs are supposed to express all the components of monoamine signaling, their occurrence has been demonstrated only in few species, which should be taken into account.

Moreover, the outcome measures chosen as inclusion criteria do not consider mechanisms of action of antidepressants other than interference with the monoaminergic system.

However, it is still unclear how antidepressants work even in humans. I suggest to take into account, or at least mention, other mechanisms of action, in order to better clarify the focus of this review protocol.

See for example Kornhuber, J.; Gulbins, E. New Molecular Targets for Antidepressant Drugs. Pharmaceuticals 2021, 14, 894. https://doi.org/10.3390/ph14090894 and refs. therein.

271 : Potential effect modifiers:

Sex, reproductive strategy, reproductive stage are not considered. This should be justified, since these biological variables can strongly influence the response to antidepressants in different molluscan species.

Reviewer #2: The manuscript by Imiuwa et al describes the methodology used by the authors to perform a systematic review of the literature on the effects of antidepressants on aquatic molluscs. If performing a systematic review on this topic could be of interest, allowing to highlight knowledge gaps (as the authors mention in their introduction), I do not understand the aim/interest of the present manuscript. Only the methodology of the review is described, no results, no discussion. There is nothing special or any novelty about the methodology used, it is the standard methodology of a meta-analysis. Moreover, it seems that at the end the systematic review will become a narrative review as the extracted data were not amenable. Thus, why not present this review directly?

Other comments:

-As the difference between “narrative review” and “systematic review” may not be clear to every reader, add few words to explain the difference.

-If target-proteins are well defined in humans, as the author said this is not perfectly understood in molluscs. So, what are “target-specific effects” and their links to the selected outcomes? Some explanation come l.231-235, but it may be a bit late.

-Explain the chosen keywords. Why do you target specific compounds and not stay at the family level using SSRIs, TCAs, SNRIs, etc. (as described in the introduction)? Or both specific compounds and families? No active metabolites were chosen as keywords while they are mentioned in the PECO question, why?

-Table 1. Remove capital letters and use only lower-case letters.

-As metabolites are also compounds of interest, it is not clear why studies on biotransformation were excluded. CYP450 proteins can also be defined as a target of drugs.

-L281. Why did the author consider the data as not amenable?

Reviewer #3: Report on the systematic review protocol proposed by Imiuwa et al. based on the criteria proposed by Plos One in Guidelines for reviewers (in italics)

Is the rationale for the proposed study clear and valid?

The review proposed on antidepressant drug pollution effects on molluscs meets the interest of scientists involved in the field of emerging contaminants in the aquatic environment. They may be lab researchers, depuration plant engineers, as well as regulators focused on prioritization to counteract environmental pollution.

Evidence mainly dating back in 2014-15 suggested that molluscs are much more vulnerable to the effects of antidepressants with respect to animals from other phyla. In the latest years, studies and results were more fragmented since addressed to many parameters and obtained in different animal models, thus a systematic review is advisable to reach a wide and updated vision.

Is the protocol technically sound? Will it effectively achieve its aims, and test the stated hypotheses?

I agree with the point of view presented by the authors on the absence or different function of enzymes in molluscs versus vertebrates (the example at line 120-123 and later on), and I support their willing to go in-depth on this point. Too many papers claim for different function alterations while referring only to different gene expressions, but it is not always the case.

Also I agree with the search on the three data bases reported in Table 1.

At page 11, lines 225-226 active metabolites of antidepressants are mentioned, but they are not in the list of Table 1.

As to the outcome, although the in vitro studies are excluded from the review, I would extend the article screening also to “cell physiology” not only neurophysiology since many effects regard cell receptor interaction, modulation of cell also signaling, organelles function, etc. In bivalve early life stages in vivo studies are available on antidepressants that are not addressed strictly to as development.

Is the methodology feasible and detailed enough to make the work replicable?

How will the authors identify and select the “grey” literature? This is not mentioned, however I suggest doing it only from official Reports and from Thesis that are hopefully validated by supervisors at specific Universities following the same criteria proposed in Table 2 and 3.

7. PLOS authors have the option to publish the peer review history of their article (what does this mean?). If published, this will include your full peer review and any attached files.

Reviewer #1: No

Reviewer #2: No

Reviewer #3: No

---

## [Author Response · Author response to Decision Letter 0]

23 May 2023

Included as an attachment (Response to reviewers)

---

## [Editor Report · Decision Letter 1]

8 Jun 2023

Understanding target-specific effects of antidepressant drug pollution on molluscs: a systematic review protocol

PONE-D-23-08500R1

Dear Dr. IMIUWA,

We’re pleased to inform you that your manuscript has been judged scientifically suitable for publication and will be formally accepted for publication once it meets all outstanding technical requirements.

Kind regards,

Peter P. Fong, Ph.D

Academic Editor

PLOS ONE
---

## [Editor Report · Acceptance letter]

19 Jun 2023

PONE-D-23-08500R1 

Understanding target-specific effects of antidepressant drug pollution on molluscs: a systematic review protocol 

Dear Dr. IMIUWA:

I'm pleased to inform you that your manuscript has been deemed suitable for publication in PLOS ONE. Congratulations! Your manuscript is now with our production department. 

Kind regards, 

on behalf of

Dr. Peter P. Fong 

Academic Editor

PLOS ONE